# Surgical Training on Ex Vivo Ovine Model in Otolaryngology Head and Neck Surgery: A Comprehensive Review

**DOI:** 10.3390/ijerph19063657

**Published:** 2022-03-19

**Authors:** Matteo Fermi, Francesco Chiari, Francesco Mattioli, Marco Bonali, Giulia Molinari, Matteo Alicandri-Ciufelli, Lukas Anschuetz, Ignacio Javier Fernandez, Livio Presutti

**Affiliations:** 1Department of Otorhinolaryngology Head and Neck Surgery, IRCCS Azienda Ospedaliero-Universitaria di Bologna, Policlinico S.Orsola-Malpighi, 40138 Bologna, Italy; matteo.fermi.med@gmail.com (M.F.); dr.giuliamolinari@gmail.com (G.M.); ignafernandez@yahoo.it (I.J.F.); livio.presutti@unibo.it (L.P.); 2Department of Specialist, Diagnostic and Experimental Medicine (DIMES), Alma Mater Studiorum, Università di Bologna, 40138 Bologna, Italy; 3Department of Otorhinolaryngology Head and Neck Surgery, University Hospital of Modena, 41125 Modena, Italy; francesco.mattioli@unimore.it (F.M.); bonamed1984@hotmail.it (M.B.); matteo.alicandriciufelli@unimore.it (M.A.-C.); 4Department of Otorhinolaryngology, Head and Neck Surgery, Inselspital, Bern University Hospital, University of Bern, 3010 Bern, Switzerland; lukas.anschuetz@insel.ch

**Keywords:** ex-vivo ovine model, surgical education, surgical training, head and neck surgery, endoscopic ear surgery, laryngotracheal surgery, salivary gland surgery, pediatric endoscopy, oculoplastic and orbital surgery

## Abstract

*Background:* Nowadays, head and neck surgical approaches need an increased level of anatomical knowledge and practical skills; therefore, the related learning curve is both flat and long. On such procedures, surgeons must decrease operating time as much as possible to reduce the time of general anesthesia and related stress factors for patients. Consequently, little time can be dedicated for training skills of students and young residents in the operating theater. Fresh human cadavers offer the most obvious surrogate for living patients, but they have several limitations, such as cost, availability, and local regulations. Recently, the feasibility of using ex vivo animal models, in particular ovine ones, have been considered as high-fidelity alternatives to cadaveric specimens. *Methods*: This comprehensive review explores all of head and neck otolaryngology applications with this sample. We analyzed studies about ear surgery, orbital procedures, parotid gland and facial nerve reanimation, open laryngeal and tracheal surgery, microlaryngoscopy procedures, laryngotracheal stenosis treatment, and diagnostic/operative pediatric endoscopy. For each different procedure, we underline the main applications, similarities, and limitations to human procedures so as to improve the knowledge of this model as a useful tool for surgical training. *Results*: An ovine model is easily available and relatively inexpensive, it has no limitations associated with religious or animal ethical issues, and it is reliable for head and neck surgery due to similar consistencies tissues and neurovascular structures with respect to humans. However, some other issues should be considered, such as differences about some anatomical features, the risk of zoonotic diseases, and the absence of bleeding during training. *Conclusion*: This comprehensive review highlights the potentials of an ex vivo ovine model and aims to stimulate the scientific and academic community to further develop it for other applications in surgical education.

## 1. Introduction

Skills development in otolaryngology head and neck surgery (OHNS) is a real challenge during resident training. In fact, surgical procedures are becoming increasingly sophisticated due to the advent of new technologies (i.e., microscopic, endoscopic, exoscopic, and robotic surgery) and the intrinsic complexity of the anatomic district. The practical teacher–learner relationship takes places exclusively on the operating field. However, during routine surgery, the expert surgeon is frequently asked to reduce the operating time either for clinical or organizational purposes, and the time left for young residents’ surgical education is limited [1]. Currently, for the above-mentioned reasons, most OHNS trainees still have limited exposure to surgical procedures during their training, which further slows down the uptake of practical skills. Indeed, a training model is needed to both allow students to acquire practical abilities and to enable teachers to document technical competence in a standardized fashion [2].

Cadaveric surgical dissection remains the gold standard for surgical education, but high costs and limited availability make it prohibitively inaccessible for regular practice, especially in several countries, due to regulatory issues. Synthetic specimens have been developed to allow the simulation of several surgical procedure. However, none of them has reached significant consensus in the OHNS community, and the costs are often still considerable.

The ex vivo animal model has been tested to simulate several OHNS procedure to overcome some of those limitations. In particular, the ex-vivo ovine model has been thoroughly described in this anatomic district and has shown its versatility in the simulation of OHNS. Moreover, it is broadly available and relatively inexpensive, and it has no limitations associated with religious or animal ethical issues. This model is reliable for OHNS due to a similar texture of skin, subcutaneous tissues, muscles, and bone compared to human ones. The anatomy of the head and neck district is similar, and several surgically relevant anatomic relationships are preserved.

However, there are also several disadvantages. The risk of a zoonotic disease might be relevant considering handling of fresh ovine tissue. Even limited exposure to *C. burnetii* can lead to the development of Query fever (Q fever), a treatable but potentially dangerous disease [3]. The major route for human contamination is aerosolized amniotic and placenta materials [4]. To prevent this risk, animals can be tested for Q fever using serological methods. Selected tissues from male or prepubescent female sheep should reduce the risk for *C. burnetii* exposure [5]. Another limitation of such model is the absence of bleeding, whose intraoperative control is essential, and the development of techniques capable to rescue such event is of utmost importance [6].

In particular, the ex vivo ovine model showed its utility in endoscopic ear surgery [7,8,9,10], on orbital procedures [11,12], on parotid gland surgery, and in facial nerve reconstruction/reanimation [13,14]. In addition, it is also used in open laryngeal and tracheal surgery training [15,16,17], in microlaryngeal surgery, and lastly, to simulate several pediatric procedures [18].

The aim of this paper is to provide a comprehensive review about the advantages and disadvantages of the ex vivo ovine model in the surgical training of the OHNS field.

## 2. Ovine Model for Endoscopic Ear Surgery Training

The cadaveric temporal bone is the gold standard for simulation in otologic surgery. However, the ex vivo ovine model has shown its validity for the endoscopic transcanal surgical training. The feasibility of different surgical steps has been explored by several authors.

Comparative anatomy studies highlighted the similarities and differences between the sheep and the human middle ear [19]. Moreover, the eligibility of the sheep model was confirmed by compared radiological studies [20].

Herein are the most relevant peculiarities of the ovine’s ear:(a)The tympanic membrane (TM) has a large pars flaccida posteriorly and superiorly to the malleus, covering the epitympanic space;(b)There is not a fibrous or bony tympanic annulus;(c)The malleus lies more anteriorly and has a longer handle;(d)The incus has a shape like human, but the processes are inverted (the short process corresponds to the human’s long one);(e)The stapes is smaller, and the two crurae are thicker, and they are closer together;(f)The facial nerve is always dehiscent;(g)The retrotympanum, containing the round window niche, is hidden behind a bony prominence of the EAC, and it is not pneumatized.

Okhovat et al. compared the feasibility of endoscopic transcanal tympanoplasty on the ovine model and synthetic temporal bone, showing how the first was a significantly more realistic simulation tool than the latter [10].

Anschuetz et al. [7] developed a surgical training program for endoscopic canaloplasty, myringoplasty, and ossiculoplasty. Canaloplasty was often required to widen the ovine’s external auditory canal (EAC) and achieve an adequate view on the TM. Tympanomeatal flap could be performed, being careful not to tear it due to its thickness and absence of a fibrous annulus. Myringoplasty was performed by placing an artificial membrane or a piece of ovine’s septal cartilage and perichondrium in underlay fashion. Ossiculoplasty could be performed with remodeled incus or malleus, prosthesis, or cartilage.

Fernandez et al. proposed a training program for salvage procedures in endoscopic stapes surgery using the ovine’s ear. They suggested a step-by-step building of different difficult scenarios, which can be encountered during stapes surgery (e.g., floating footplate, footplate fracture, necrosis of the long process of the incus, incudomalleolar luxation, overhanging facial nerve) [8]. In fact, the variability of situations that can be found during stapes surgery, especially in revision surgery cases, can increase the difficulty both in teaching and learning it. This model was adequate to prepare the surgeon to deal with such challenging conditions and be ready to manage them in a real-life situation.

Beckmann et al. [9] examined the feasibility of the ovine model for exclusive endoscopic laser stapedotomy. They concluded that the learning curve was significantly association between the operative time and occurring intraoperative complications, globally indicating its usefulness.

Lastly, the feasibility of a broad spectrum of endoscopic approaches on the ex vivo ovine model was described by Bonali et al., extending the dissection to the internal auditory canal and the round window (Figure 1) [21].

Unfortunately, the sheep is not suitable for transmastoid surgery due to the scarce development of the mastoid compartment and the important anatomic differences with the human.

Future perspective might be to expand the possibility of the model for lateral skull base surgery, trying to investigate the feasibility of posterior cranial fossa and middle cranial fossa approaches.

## 3. Ovine Model for Orbital Approaches

Fresh animal models are widely used for teaching intraocular surgery in OHNS and ophthalmology training. Isaacson et al. [11] described the suitability of a sheep model for teaching oculoplastic surgery. They demonstrated that it can be used to train blepharoplasty and ptosis repair, lateral canthotomy and inferior limb cantholisis, tranconjunctival approaches, and medial rectus resection or recession taking advantage of similar anatomy and tissue quality.

However, the ovine’s orbit is different compared to human on, and herein, we report some of the main differences [11]:(a)The sheep orbital cavity is deeper with a sharply angled and complete rim;(b)The sheep has an incomplete orbital floor and lateral walls, with a large opening that communicates with the temporal fossa and coronoid process of the mandible;(c)There is some orbital fat deep in the orbital septa of both lids, which does not protrude easily when the orbital septum is incised;(d)A sheep’s tarsus is thinner centrally compared to humans’;(e)The sheep extraocular muscles are less robust;(f)The sheep eyelids are thinner;(g)The lachrymal system is similar, but it has different proportion: single puncta are located at the anterior lid margins communicating with a rudimentary lacrimal sac; the nasolacrimal duct extends several centimeters along the long ovine snout. Thus, this model is inappropriate for endoscopic dacryocystorhinostomy training [12].

Despite all of differences reported, the sheep is considered a valid model for these procedures.

## 4. Ovine Model for Parotidectomy and Facial Nerve Reanimation Training

Facial nerve damage is considered the most important complication in parotid gland surgery. It is mandatory to have a good knowledge of anatomical landmarks and good skills to perform reconstruction or reanimation of facial nerve. Surgical ability in macroscopic and microscopic fields are required for such surgery. Macroscopical steps are based on the identification and dissection of anatomical structures. Microscopical ones require precise surgical maneuvers, such as during anastomosis. It is a common opinion that microscopic surgery requires a great deal of time in practice, and it can be useful to achieve experience. Niimi et al. [22] emphasized how the facial nerve reconstruction and regeneration are feasible on the ex vivo ovine model, especially in terms of elastic resistance and consistence, which allows to closely simulate the forces during the incision or dissection as well as the delicate gesture to use in contact with vascular or nervous structures. Milner et al. [14] compared the ovine and the porcine model in parotid surgery. The first one is highly realistic regarding anatomical structures with similar landmarks to humans. This model is deemed extremely useful for teaching anatomy and surgical planning.

Some differences are found between human and ovine parotid gland, such as:(a)The ovine skin is thicker than humans’;(b)The tragal pointer is bigger than a human one;(c)The ovine zygomatic auricular muscle is found instead of the human anterior auricular muscle;(d)The ovine temporal artery is found instead of the human stylomastoid artery;(e)The ovine facial main trunk is shorter and divided in three branches.

Few human landmarks can be used on the ovine model. The ovine facial nerve is identified following the EAC, but no relationship with the posterior belly of the digastric muscle is found, while the pointer and the tympanomastoid suture are still reliable surgical landmarks.

Ghirelli et al. [13] made a structured training program based on a step-by-step dissection divided into macroscopic and microscopic items. The macroscopic procedure starts with a modified Blair’s incision. The parotid gland is reached on a subsuperficial muscular aponeurotic system, and the superficial landmarks are found. The zygomatic auricular muscle is incised close to tragus to access the superior part of the gland. The following step consists of skeletonization of the EAC to expose the cartilaginous pointer. The main trunk of the facial nerve is found below the pointer and the temporal artery. A superficial parotidectomy with dissection of peripheral facial nerve branches is performed, followed by the parotid’s deep lobe removal (i.e., total parotidectomy). To prepare for the microscopic surgical exercises, the main trunk of facial nerve is sectioned near the stylomastoid foramen, while the hypoglossal nerve is followed as deeply as possible, and then, it is sectioned itself. With a microscopic approach the two nerves are released and anastomosed in an end-to-end fashion (Figure 2). Finally, the marginal mandibular branch is sectioned in the middle of its course and anastomosed in an end-to-end fashion.

## 5. Ovine Model for Open Laryngeal Surgery Training

Different open surgical procedures are available to treat laryngeal cancers, such as total laryngectomy and subtotal laryngectomies, also referred to as open partial horizontal laryngectomy (OPHL) [23]. This classification is based on the craniocaudal extent of resection, and it aims at giving a common name to slightly different procedures. According to such taxonomy, three types of OPHL are defined: type I (horizontal supra-glottic laryngectomy), type II (supracricoid laryngectomy), and type III (supratracheal laryngectomy). Suffixes “a” and “b” are used in type II and III OPHLs to, respectively, reflect the sparing or not sparing of the suprahyoid epiglottis. Various extensions to one arytenoid (ARY), base of tongue (BOT), piriform sinus (PIR), and crico-arytenoid unit (CAU) are indicated in the case of macroscopic extension to these structures.

The complexity of open laryngeal surgery requires a long learning curve, especially the OPHLs, where the surgeon must be able to expand the procedure as far as needed on a case-by-case basis. Surgeons should perform the best and most conservative surgical approach according to the tumor extension and patient’s features. For such reason, open laryngeal surgery training on the ovine model has been developed, showing the capability to perform every single surgical step, both for OPHLs and total laryngectomy, as in the human.

The anatomical structures of sheep are like humans. However, there are some differences reported [24], such as:(a)The thyroid cartilage is developed on a craniocaudal extent, and the anterior edge formed by the two laminae appears more rounded;(b)The cricoid cartilage is fairly similar to human apart from its inconstant process at the inferior margin of the lamina;(c)The arytenoid cartilage has a bigger size, especially at its vocal process;(d)The laryngeal ventricle is absent.

Fermi et al. developed and validated an ex vivo ovine model suitable for open laryngeal surgery, such as total laryngectomy and OPHLs. Surgical simulation on this model allows young residents to better understand the surgical aspects of related procedures. Moreover, experienced laryngologist might use the animal sample to effectively warm up before a planned surgery where a complicated extension or reconstruction is required [15].

## 6. Ovine Model for Suspension Microlaryngoscopy Training

The suspension microlaryngoscopy is an important procedure in OHNS training program. This technique allows to reach laryngeal structures through a transoral approach by using a laryngoscope, which is used to expose the laryngeal area and then suspended to allow the surgeon to work with both hands on the pharyngolaryngeal structures. A surgical microscope is used to provide magnification and to deliver a laser beam. Through this approach, the surgeon executes diagnostic and operative procedures, such as the research of dysphonia etiology, resection of suspected or known malignancies, and phonosurgical interventions for vocal fold paralysis (i.e., injective laryngoplasty, glottoplasty, lateralization of vocal fold, posterior cordectomy, etc.).

This is a routine and diffuse technique, and it may also be considered one of the first surgical skill to be achieved by young trainees.

The ovine model is a viable option to replicate human anatomical features. The main differences reported as follows:(a)The elongated shape of the ovine jaw compared to the human one;(b)A long and high-riding epiglottis;(c)Prominent arytenoids with difficult-to-access vocal folds;(d)The lack of vocal ligament;(e)A shorter distance from the oropharynx to the larynx.

Isaacson et al. [16] firstly used the ovine model for teaching suspension laryngoscopy. During this approach, several procedures can be realized, such as injection laryngoplasty, hydrodissection and incision, endolaryngeal suturing, and cordectomies. Tan et al. [25] described in their paper another similar experience.

Despite variations in proportion and anatomical features, the sheep provides an inexpensive and a safe model to develop suspension laryngoscopy and basic phonosurgical skills. It might also be employed for transoral laser microsurgery using dedicated devices, such the *larynx-box* described by Mattioli et al. [26].

## 7. Ovine Model for Laryngotracheal Surgery

Laryngotracheal stenosis (LTS) is a complex disease potentially causing devastating consequences, including respiratory failure and death. Several etiologies have been identified, including iatrogenic (i.e., the result of a severe trauma related to endotracheal intubation due to the direct compression of the tube to the posterior glottis mucosa, which progresses to fibrosis and scar, leading to stenosis), neoplastic, autoimmune, infectious, traumatic, and idiopathic [27].

The LTS management is complicated, especially considering the variety of surgical treatment, which should be mastered to choose the best option possible. In such disease, especially, the best chance lies in the first surgical attempt. Thus, a wide knowledge of all laryngotracheal surgeries, both in endoscopic [28] and in open approaches, would be needed [17,29].

A training model may be a valid option to improve surgical skills. Rangus et al. [17] described the sheep as a realistic surgical simulator for pediatric laryngotracheal reconstruction (LTR), which is indicated for grade I and grade II subglottic stenosis, according to the Cotton–Meier classification. At the same time, Nisa et al. [29] considered lamb tissue a good model to train for partial crico-tracheal resection (PCTR), which is indicated for grade III and IV subglottic stenosis and as salvage surgery after failed LTR. Ghidini et al. [30] described some other procedures, such as tracheostomy, slide tracheoplasty, tracheal resection-anastomosis, and main endoscopic techniques (i.e., posterior cordectomy, laryngeal cleft repair).

Herein, some differences between the ovine’s and human’s laryngotracheal frame are reported [29]:(a)The ovine hyoid bone has a larger minor cornu extending laterally and superiorly;(b)The anterior commissure is located at the lower edge of the thyroid cartilage;(c)The arytenoids are comma shaped, and interarytenoid space is reduced due to a thin interarytenoid muscle;(d)The laryngeal Morgagni’s ventricle is lacking in the ovine larynx;(e)The ovine’s tracheal rings are near-complete and have a very small membranous trachea.

However, on average, size, shape, and tissue handling, the features of ovine’s airway are closely similar to a pediatric one for the purpose of simulating laryngotracheal reconstruction and offering the visual and tactile feedback to learn new skills (Figure 3). Given the complexity of pediatric airway surgery, there is a significant benefit in simulation based on ex vivo ovine model.

## 8. Ovine Model for Pediatric Flexible Endoscopy Training

Due to small airways and poorly modulated protective reflex with a high risk of laryngospasm, bronchospasm, and bradycardia, neonates and infants cannot be considered valid specimens to learn pediatric flexible endoscopy. However, it is sometimes needed to operate in emergency situations, and thus, it is mandatory to develop such abilities as fast as possible. The ex vivo ovine model enables trainees to improve their learning curve. The ovine laryngotracheal complex offers an attractive anatomical model for simulation of pediatric flexible endoscopy. The size, shape, and tissue-handling feature of the sheep airway is like the human pediatric airway, so it offers the visual and tactile feedback necessary to learn this practical skill. Herein, some differences between the ovine’s and infant’s airway framework are reported [31]:(a)The sheep’s head has a long nose and a short nasopharynx;(b)The ovine’s larynx is high riding, and the epiglottis interdigitates with a soft palate;(c)There are no false vocal folds or laryngeal ventricles.

Isaacson et al. [18] used the ex vivo ovine model for pediatric flexible endoscopy training. Despite variations in proportion and structure, the experience in a sheep airway was like pediatric endoscopy. For example, transnasal intubation, navigating toward the laryngeal vestibule through a flexible endoscopic guidance, has the look and feel of the human pediatric procedure, supporting its use for anesthesiology and OHNS residents in training.

## 9. Conclusions

The ex vivo ovine model could be considered a valid option to increase surgical skills of otolaryngology head and neck surgery trainees. Both simple and complex procedures can be performed on this model. Moreover, both microscopic and endoscopic procedures might be trained on the ex vivo ovine model.

Advantages of this model are its versatility in the simulation of OHNS due to similar consistencies tissues and neurovascular structures with respect to humans; it is also available and inexpensive. It has no limitations associated with religious or animal ethical issues. However, some other issues should be considered, such as differences about some anatomical features, the risk of zoonotic diseases, and the absence of bleeding during training. The risk of zoonotic disease can be highly reduced by screening fresh ovine tissue for Q fever antibodies and by wearing all the proper personal protection equipment. Moreover, future studies are required to validate such a model for in vivo surgical training to take advantage of intraoperative bleeding.

This comprehensive review highlights the potentials of this model and aims to stimulate the scientific and academic community to further develop it. A future perspective might be the development of a lateral skull-base training program and the application of the new exoscopic technology.

## Figures and Tables

**Figure 1 ijerph-19-03657-f001:**
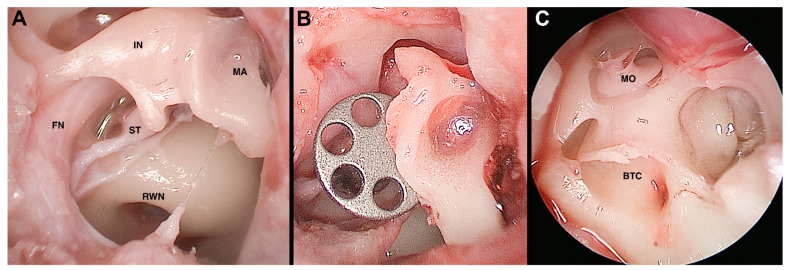
Ovine model, endoscopic view, right side. Panel (**A**): Middle ear cleft anatomy; Panel (**B**): Partial ossicular chain reconstruction with titanium prosthesis; Panel (**C**): Vestibule and cochlea opening. IN, incus; MA, malleus; ST, stapes; FN, facial nerve; RWN, round window niche; BTC, basal turn of the cochlea; MO, modiolus.

**Figure 2 ijerph-19-03657-f002:**
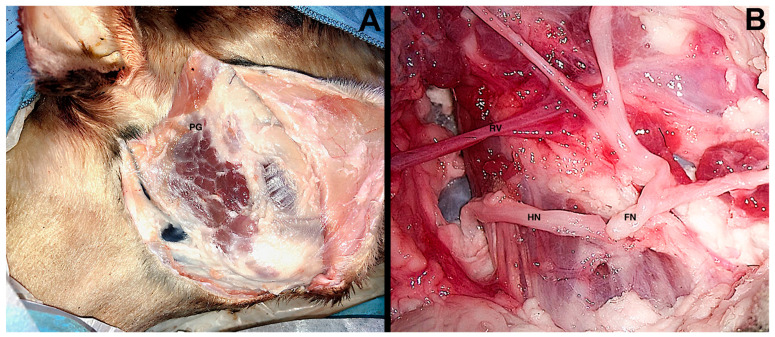
Ovine model, right side. Panel (**A**): Subsuperficial muscular aponeurotic elevation and parotid gland exposure; Panel (**B**): Hypoglossal-facial end-to-end anastomosis. PG, parotid gland; HN, hypoglossal nerve; FN, facial nerve; RV, retromandibular vein.

**Figure 3 ijerph-19-03657-f003:**
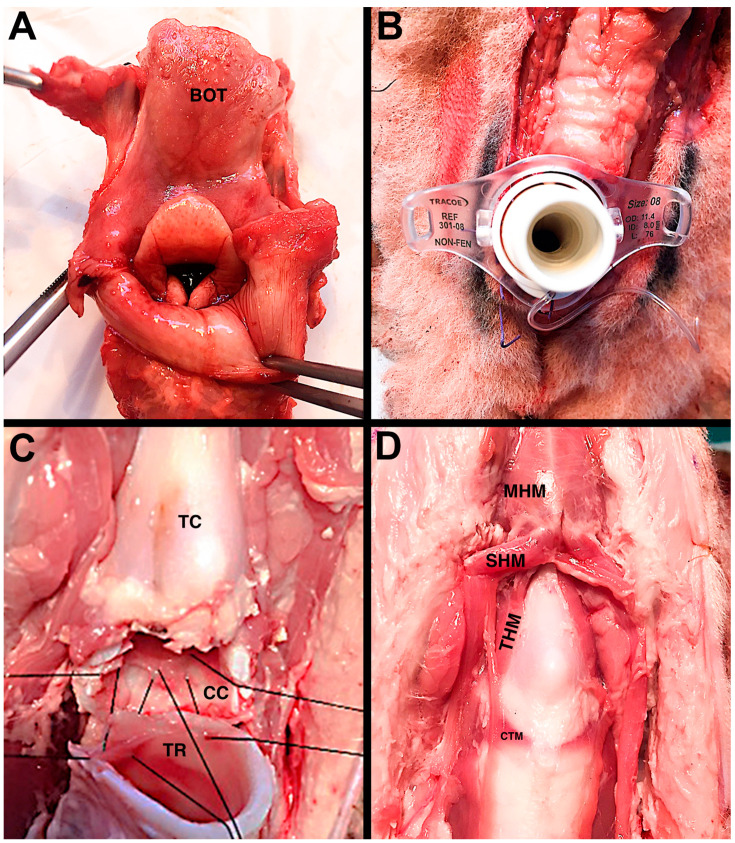
Ovine model. Panel (**A**): Visualization of the laryngeal vestibule; Panel (**B**): tracheostomy with tracheal cannula in place; Panel (**C**): Partial cricotracheal resection (PCTR), posterior anastomosis; Panel (**D**): Frontal view of the laryngotracheal framework. BOT base of tongue; TC, thyroid cartilage; CC, cricoid cartilage; TR, trachea; MHM, mylohyoid muscle; SHM, sternohyoid muscle; THM, thyrohyoid muscle; CTM, cricothyroid muscle.

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
