# Peer review of "Surgical Training on Ex Vivo Ovine Model in Otolaryngology Head and Neck Surgery: A Comprehensive Review"

_ijerph, 2022, doi:10.3390/ijerph19063657_

Round 1

Reviewer 1 Report

Dear Authors,

I appreciate your efforts in this review study .

I found that this study is  an interesting  review with aim of  improving  the knowledge of this model as a useful tool for surgical training and education.

Please find below comments: 

  • The aim of the review study is mentioned in the background of the abstract.
  • it is better to explain the  general advantages and disadvantages of the ovine model based on this review  in the conclusion.
  • I think that researchers should leave their comments on how to improve this model's limitations in the conclusion part.
  • Also, it is better to revise keywords ( for example , in this part did not mention the ovine model.

Reviewer 2 Report

The authors' review article covers comprehensive knowledge of ovine ex-vivo model for training of head and neck surgery. They digested well the previous reports about practice using ovine model for ear, orbital, parotid gland, laryngeal, and tracheal surgery, including pediatric flexible endoscopy. Their article is really useful for head and neck surgeons to understand the benefit of ovine model for younger doctors' education.

Some minor issues should be reconsidered.

A couple of pictures or illustrations should be added or cited for readers' deeper understanding.

Line 29 and 61, "it has not limitations associated with religious,..." should be "It has no limitation  associated..."

Line 97, "the is smaller..."     ?

Line 98, "like the human one;"   " ; " is not necessary.

Line 157 should be the subheading.

Line 179, 247, and 304, periods are not necessary in accordance with other itemization?

Line 324-, the references are not written in the style of the journal.

Reviewer 3 Report

The paper is interesting and well written. Some minor corrections to improve the overall quality:

There is a typo in the title. Please correct

Introduction

  • please apply the PRISMA statement and the picot framework
  • add a flow diagram of the revision process
  • modify et AL. in ''et al.''
  • please add figures of the training model.
  • report the citation as [15] at the end of the sentence.
  • in the different paragraph don't repeat the  ovide model always.
  • in the conclusions always use could, may.. not stating a concept but proposing.
